## [Transparent Peer Review file · Nature Communications]

Bioinspired maskless structural colour patterning via tunable nanoparticle segregation

Corresponding Author: Professor Ming Xiao

Version 0:

Reviewer comments:

Reviewer #1

(Remarks to the Author)

General Comment: This manuscript presents a novel approach for structural color patterning via tunable nanoparticle segregation. The authors disperse the silica nanoparticles in acrylic resin. Driven by interfacial oxygen inhibition, the silica nanoparticles can migrate toward oxygen-permeable substrates, forming the nanoparticle-enriched disordered layer. By controlling the local segregation thickness, high-resolution structural colour patterns can be created. The authors apply the grayscale digital light processing printing to customize the structural-color images and graphics, which are promising for visual display and information encryption. Furthermore, they demonstrate the infrared camouflage based on the selective mid-infrared reflectivity. The integration of digital printing and tunable nanoparticle segregation shows technical maturity and creativity, yet several critical points require further clarification.

1. Title: The term “programmable” literally means “capable of being programmed” and typically implies the reciprocating cyclicality of actions. However, in this paper, the aggregation of nanoparticles is unidirectional and lacks recyclable cyclicality. Therefore, using the term “programmable” would lead to ambiguity. It is recommended to replace it with other appropriate terms, such as “tunable” or “customized”, which merely need to indicate the tunability of aggregation thickness.

2. Introduction: Background literature is insufficiently comprehensive. Current research on structural colors is extensive. Firstly, structural colors can be fabricated not only via lithographic methods but also through additive approaches like self-assembly. Additionally, the color-generating structures are not limited to nanoscale. Microscale TIR structures can also generate intense structural colors. Furthermore, for structural-color patterning, 3D printing is only a minor representative. Compared with other techniques like inkjet printing, 3D printing offers lower degrees of freedom, precision, and customization capability. Therefore, regarding the research background, the authors are suggested to provide further supplements that are not be limited to 3D printing.

3. Main text: Is it possible to provide a precise quantitative definition of disordered photonic structures, i.e., the range distinguishing ordered and disordered self-assembled structures? Furthermore, the unique optical properties brought by disordered photonic structures should be emphasized. While the authors have noted that ordered and disordered structures exhibit differences in iridescent properties in the word descriptions, this distinction is not reflected in the figures of the main text. It is suggested to supplement this content accordingly.

4. Fig. 1c: From this schematic diagram, it is hard to understand the differences in polymerization degree between the upper and lower regions. If the authors aim to demonstrate the influence of the substrate on polymerization degree, this aspect should be prominently emphasized.

5. Fig. 4b: Generally speaking, the color of ordered photonic structures is brighter than that of disordered ones. However, based on the results, the color intensity of ordered photonic structures is instead weaker. Furthermore, as the nanoparticle size increases, the color of the ordered surfaces does not show any change. What is the reason for this phenomenon?

6. Fig. 4: The term “hierarchical” typically refers to multi-scale hierarchical changes ranging from the nanoscale to the microscale. Obviously, it is inappropriate in this system, as the optical effect-generating system relies primarily on nanostructures and does not involve hierarchical characteristics. It is recommended to replace it with other appropriate terms.

Reviewer #2

(Remarks to the Author)

This work presents a new method to achieve structural color patterns via tunable nanoparticle segregation. During

photocuring, silica nanoparticles dispersed in acrylic resin migrate toward oxygen-permeable substrates, forming a nanoparticle-enriched disordered layer. Such segregation is driven by interfacial oxygen inhibition and kinetically governed by the photocuring rate. As a result, by employing the grayscale digital light processing printing, programmably controlling the local segregation thickness to create pixelized structural color patterns is demonstrated. In the meantime, some potential applications such as visual display, information encryption, and infrared camouflage are illustrated.

However, structural colors are not new, with many examples in recent years to realize vivid color patterns. Though the method proposed in this research can be potentially used for structural colors, the really awesome results are still not demonstrated, just taking the bioinspired bird feathers of grassquit for an example. Maybe, some other fantastic applications or further optimizations for structural colors need to be exploited.

On the other hand, some technical issues are also remaining to be addressed:

1. Though the segregation of silica nanoparticles depends on the conditions like the substrate material, oxygen addition, and light intensity, the self-assembly process is still the most important. How to get the vivid color patterns simply, effectively and repetitively is the basic question that needs to be answered. It seems more like an engineering aspect.
2. In line 252, "As shown in Fig. 3a" should be "As shown in Fig. 4a". How to define the "highly saturated colour"? And also, in line 268, how to define the "vivid colouration"? In Fig. 4b, the change of colors according to the increasing volume fraction is not so obvious, especially for the particle size below 194nm.
3. After the line 287, photonic ink containing different percentage of silica nanoparticles and the size of nanoparticles changes very often, making a common understanding of how to select the specific configuration of operations difficult to follow.
4. For the infrared camouflage, infrared imaging seems in the LWIR waveband. So, what is the purpose and usage of the related part for the MWIR? It should be given in a more explicit way to show the relationship.

Reviewer #3

(Remarks to the Author)

This manuscript suggests a one-step, mask-free strategy for high-resolution structural color patterning by exploiting programmable nanoparticle segregation during photocuring. Silica nanoparticles dispersed in acrylic resin migrate toward oxygen-permeable substrates, forming a disordered, nanoparticle(NP)-enriched layer driven by interfacial oxygen inhibition. It was revealed that during each layer's curing using grayscale digital light processing (DLP), three distinct zones are formed, and the thickness of the NP-enriched layer is primarily determined by the photocuring rate. The thin-film interference occurring at the distinct zones enables infrared reflectivity. The meticulous elucidation of NP segregation at an oxygen-permeable interface is not trivial and holds value as a scientific discovery, but several necessary scientific descriptions are missing. I propose the following content for revision:

1. When patterned photocuring occurs at an oxygen-permeable interface, lateral oxygen diffusion (also known as reaction diffusion) parallel to the substrate is known to occur, leading to the formation of a gradient microstructure. Considering this, how are the different zones formed in the lateral direction at the boundary of different doses?
2. The principle by which the particles undergo crystallization (ordering) within the acrylate resin is not described. There should be an introductory mention of crystallization driven by interparticle repulsion from the overlap of solvation layers.
3. Without this explanation, the subsequent mention of other particles like polystyrene (PS) and ZnS is problematic, as PS typically does not crystallize in acrylate resins and has only been reported to achieve ordering within a very small processing window via entirely different mechanisms, yet this distinction is not addressed. Additionally, PS particles must be swollen by acrylic resins and there must be no interparticle repulsions in the PS system. Therefore, their migration and dispersion in the upper layer are not likely to occur in the same way.
4. Can 3D architectures with different structural colors be printed with the same silica nanoparticles by controlling the photocuring rate? Fig. 4d doesn't imply it's possible, but Fig. 5c may suggest that it is feasible.
5. The illustration of the illumination setting in Fig. 5b shows specular reflection. However, considering that all the panda images are in the same size, the observation might have occurred in the normal direction, which is a setting of diffusive scattering, not specular reflection. If the observation was done in the specular reflection, the colors from the photonic glasses should also change anyway.
6. In Figure 4d, 15 vol% was seemingly used. Calculating with the 154 nm particles, the structural coloration from the ordered non-close-packed array should result in a red color, likely in the low to mid 600 nm range, although this depends on the acrylic resin's refractive index. However, red is barely visible across the entire irradiance spectrum. Why is this the case?
7. The possibility of infra-red camouflage is interpreted as arising from thin-film interference, but the description is insufficient. It appears on line 323 of page 14 without a transition or explanation relating it to different coloration mechanisms.
8. Is there a possibility that the acrylic resin used in this research could penetrate and cause swelling of the oxygen-permeable substrate? Can the monomer flux into the polymer film also induce an additional extent of particle segregation at the interface on top of the currently discussed mechanism?

9. Can you specify whether MWIR reflection comes from the segregation zone or the depleted zone, and whether LWIR reflection comes from the depleted zone or the uniform zone?

Version 1:

Reviewer comments:

Reviewer #1

(Remarks to the Author)

The authors have thoroughly and effectively addressed all the reviewer's comments, and the revised manuscript is now significantly clearer and more accessible to readers. Their supplementary experimental demonstrations substantially enhance the scientific impact of this work. I therefore recommend the publication of this paper.

Reviewer #2

(Remarks to the Author)

Some of my concerns have been covered. However, all the engineering-related issues are still missing.

Reviewer #3

(Remarks to the Author)

I appreciate the reviewers for answering all the comments carefully. I only have comments for first to comments, Comment 3.2 and Comment 3.3. All other comments are well addressed with well-supported rationales. I sincerely appreciate the careful response from the authors.

For Comment 3.2, what I intended to ask was exactly about the gradient height of the particle segregation which was made by lateral diffusion of oxygen. Namely, I didn't mean to ask "vertical segregation layers" but the slanted height of the segregation layer. Previous papers about reaction diffusion formed cone-shaped particles by using this vertical diffusion of oxygen, which seems to happen in this case, too. I do agree that vertical segregation layer will not happen, but the slanted formation of the layer will occur. Therefore, I suppose this will also affect the resolution that is argued in this paper. I think it is worth mentioning the gradient height change of horizontal segregation layers in terms of resolution. Otherwise, readers would think there will be a sharp contrast at the boundary of strong and weak doses.

For Comment 3.3, I do understand that many papers are still arguing that the crystallization of silica particles dispersed in acrylate polymer is mainly due to electrostatic repulsion. But, many papers do not. Please refer to the paper from one of the pioneers in this silica-acrylate system, who is Dr. Peng Jiang. *J. Am. Chem. Soc.* 126, 13778-13786 (2004). Its title is "Large-scale fabrication of wafer-size colloidal crystals, macroporous polymers and nanocomposites by spin-coating." Here, in the right paragraph on page 13785, although Dr. Jiang once stated that electrostatic repulsion drove the colloidal crystallization in this system, this *JACS* paper clearly states:

1) "The interparticle electrostatic repulsion plays only a minor role due to the low dielectric constant of the ETPTA medium (3 at optical frequency)"

2) "We also tested salt effect on the colloidal crystallization process during spin coating by adding 10⁻³ M tetrabutylammonium chloride to the silica-ETPTA dispersion (10% v/v 200-proof ethanol was added as cosolvent) to screen interparticle electrostatic repulsion and make silica colloids more "hard-sphere" like. The resultant spin-coated nanocomposite exhibits the same in-plane particle separation (~1.41D) and optical transmission spectrum as those samples made without the addition of salt, indicating minor contribution of the electrostatic force to the colloidal crystallization."

It is different from salt, but even the addition of a trace amount of ethanol and water into this silica-acrylate system will destroy the crystallized state and induce particle network, which means the main driving force of repulsion is diminished by the solvents. If it would have been electrostatic repulsion, there is no reason the addition of water decrease the repulsive force but increase the attractive force. (*Adv. Mater.* 36, 2307917 (2023). Its title is "Advanced Additive Manufacturing of Structurally-Colored Architectures")

Acrylate groups form hydrogen bonds with silanol groups of silica particles and they form solvation layers. When different solvation layers overlap, disjoining pressure makes the particles repel one another, which can be also stated as steric hindrance. I think acrylate polymer is not like water, and cannot induce electrostatic repulsion strong enough to drive the whole crystallization. While water, with a high dielectric constant of 80, can facilitate ionic dissociation and induce strong electrostatic repulsion, organic media such as acrylates hinder ionization due to their low dielectric constants. Consequently, the Debye screening length becomes extremely short, making it difficult for electrostatic forces to operate over a long range. Therefore, I think the sentence that states the driving force for crystallization is MAINLY the electrostatic force is not correct, but at least it has to include steric hindrance or disjoining pressure from the overlap of solvation layers.

If the authors address these points by incorporating a brief discussion or a minor revision to the current statement regarding the crystallization mechanism, I believe the manuscript will be fully suitable for publication.

Version 2:

Reviewer comments:

Reviewer #2

(Remarks to the Author)

Most of my comments are covered in this revised version. Though this work presents a new method for nanoparticle-based structural colors, I encourage the authors to show the engineering application potentials, which will be more interested by the researchers in this field.

Reviewer #3

(Remarks to the Author)

I thank the authors to carefully revise the manuscript considering the comment. I fully agree with the publication of this manuscript.

RESPONSES TO REVIEWERS' COMMENTS FOR NCOMMS-25-91219

Reviewer #1

Comment 1.1

General Comment: This manuscript presents a novel approach for structural color patterning via tunable nanoparticle segregation. The authors disperse the silica nanoparticles in acrylic resin. Driven by interfacial oxygen inhibition, the silica nanoparticles can migrate toward oxygen-permeable substrates, forming the nanoparticle-enriched disordered layer. By controlling the local segregation thickness, high-resolution structural colour patterns can be created. The authors apply the grayscale digital light processing printing to customize the structural-color images and graphics, which are promising for visual display and information encryption. Furthermore, they demonstrate the infrared camouflage based on the selective mid-infrared reflectivity. The integration of digital printing and tunable nanoparticle segregation shows technical maturity and creativity, yet several critical points require further clarification.

Response 1.1

We thank the reviewer for the summary of this work and positive comments.

Comment 1.2

Title: The term “programmable” literally means “capable of being programmed” and typically implies the reciprocating cyclicality of actions. However, in this paper, the aggregation of nanoparticles is unidirectional and lacks recyclable cyclicality. Therefore, using the term “programmable” would lead to ambiguity. It is recommended to replace it with other appropriate terms, such as “tunable” or “customized”, which merely need to indicate the tunability of aggregation thickness.

Response 1.2

Taking the reviewer's suggestion, we have replaced the term “programmable” in title with “tunable”. Correspondingly, we have revised the relevant term in the main text by replacing “programmable” with terms such as “tunable” “customized” and “controllable” to enhance accuracy.

Comment 1.3

Introduction: Background literature is insufficiently comprehensive. Current research on structural colors is extensive. Firstly, structural colors can be fabricated not only via lithographic methods but also through additive approaches like self-assembly. Additionally, the color-generating structures are not limited to nanoscale. Microscale TIR structures can also generate intense structural colors. Furthermore, for structural-color patterning, 3D printing is only a minor representative. Compared with other techniques like inkjet printing, 3D printing offers lower degrees of freedom, precision,

and customization capability. Therefore, regarding the research background, the authors are suggested to provide further supplements that are not be limited to 3D printing.

Response 1.3

We thank the reviewer for the comment regarding the breadth of the literature. We have supplemented other strategies for making structural colour patterns in the main text, including assembly technique, two-photon printing and inkjet printing. Accordingly, we have made relevant edits to the revised manuscript:

[Page 1, Line 29-34] *“However, conventional structural colour patterning typically relies on lithography-like strategies, where unmasked regions with initial nanostructures are first photocured, followed by immobilization of masked regions with different nanostructures. To adjust nanostructures for different colours, external stimuli such as magnetic fields^{15,16}, electric fields¹⁷, temperature⁶, or solvents⁴ are applied upon curing to locally manipulate the arrangements of assembly units (nanoparticle, liquid crystal, block copolymer, etc.).”*

[Page 2, Line 39-46] *“Printing technology has been used to fabricate structurally coloured materials with high flexibility and customization^{18–20}. However, printing various colours often requires multiple resin formulations across methods such as inkjet printing, direct ink writing, or vat photopolymerization^{3,18,21–24}. To the best of our knowledge, only a few studies have preliminarily explored the concept of printing different colours using a single ink. Two-photon printing can locally customize the periodic nanostructures for structural colour patterns^{25,26}, but suffer from low throughput and high cost. Li et al. inkjet printed a full spectrum of structural colours by controlling the sizes of printed microdomes to tuning their total internal reflections (TIR)²⁷. These TIR based colours can only be observed from the backside of the printed samples.”*

Added references include:

Ref 15: Xu, J. et al. Magneto-responsive cellulose nanofiber hydrogels: dynamic structuring, selective light transmission, and information encoding. ACS Nano 19, 14063–14072 (2025).

Ref 20: Yang, W., Li, J., Wang, J. & Jiang, L. Bioinspired digital structural color patterning based on photonic paper/ink systems. Adv. Funct. Mater. e27923 (2025) doi:10.1002/adfm.202527923.

Ref 25: Liu, K. et al. 3D printing colloidal crystal microstructures via sacrificial-scaffold-mediated two-photon lithography. Nat. Commun. 13, 4563 (2022).

Ref 26: Wang, J. et al. Nanograting-based dynamic structural colors using heterogeneous materials. Nano-Micro Lett. 17, 59 (2025).

Ref 27: Li, K. et al. Facile full-color printing with a single transparent ink. Sci. Adv. 7, eabh1992 (2021).

Comment 1.4

Main text: Is it possible to provide a precise quantitative definition of disordered photonic structures, i.e., the range distinguishing ordered and disordered self-assembled structures? Furthermore, the

unique optical properties brought by disordered photonic structures should be emphasized. While the authors have noted that ordered and disordered structures exhibit differences in iridescent properties in the word descriptions, this distinction is not reflected in the figures of the main text. It is suggested to supplement this content accordingly.

Response 1.4

We thank the reviewer for raising questions about the degree of order and optical properties. In our systems, disordered photonic structures are isotropic, short-range correlated without long-range order. Various approaches can be used to examine the existence of long-range order, such as fast Fourier transform (FFT) analysis of SEM images (*Macromolecules* 2022, 55, 8345–8354; *Nat. Commun.* 2024, 15, 1874) and small X-ray scattering measurements (*ACS Nano* 2024, 18, 33864–33874). We have supplemented the FFT images of SEM images for the back and exposed sides in Fig. 4a. The FFT image of the back side with clear diffraction spots indicates a long-range ordered structure, while the back side shows concentric rings indicating the loss of long-range order.

To emphasize the different optical effects from the ordered and disordered structures, we have added the angle-resolved spectra and photos of the back and exposed sides of the sample in Supplementary Fig. 20. Accordingly, we have made relevant edits to the revised manuscript:

[Page 11, Line 258-266] *“As shown in Fig. 4a, the back side, which contacts the printer platform, displays bright metallic colour. Its reflectance spectrum shows a sharp peak, shifting from 605 nm to 536 nm as the measurement angle changes from 20° to 50° (Supplementary Fig. 20). This optical feature arises from the long-range ordered arrangement of nanoparticles, as further evidenced by the FFT image showing clear diffraction spots. In contrast, the exposed side, which faces the light source, exhibits a less saturated blue grey colour with nanoparticles packing in a disordered manner. Correspondingly, its FFT image presents concentric rings, indicating the loss of long-range order. Its reflectance spectrum contains a peak at 440 nm remaining almost unchanged as the measurement angle changes (Supplementary Fig. 20), consistent with the disordered arrangement.”*

Fig. 4(a). Janus optical features driven by ordered structures in the back side and disordered structures in the exposed side. The photographs (0.8×0.8 cm), schematic diagram, and SEM images containing their fast Fourier transform (FFT) images are presented, respectively. Scale bar, $1 \mu\text{m}$.

Supplementary Fig. 20. Angle-resolved specular reflectance spectra. (a) The back side and (b) the exposed side. The insets show the corresponding photos with different viewing angles. The sample is made from photonic ink containing 30 % v/v silica nanoparticles (194 nm) and 1% w/v TPO photocured under the irradiance of 2.0 mW/cm^2 .

Comment 1.5

Fig. 1c: From this schematic diagram, it is hard to understand the differences in polymerization degree between the upper and lower regions. If the authors aim to demonstrate the influence of the

substrate on polymerization degree, this aspect should be prominently emphasized.

Response 1.5

We thank the review for the comment. Fig. 1c was intended to show how we prepare samples in Fig. 1d. To avoid confusion, we have added nanoparticles to show the segregation towards the substrate in new Fig.1c.

Fig. 1(c). Schematic showing the preparation of the photonic film for verifying the segregation phenomenon on different substrates.

Comment 1.6

Fig. 4b: Generally speaking, the color of ordered photonic structures is brighter than that of disordered ones. However, based on the results, the color intensity of ordered photonic structures is instead weaker. Furthermore, as the nanoparticle size increases, the color of the ordered surfaces does not show any change. What is the reason for this phenomenon?

Response 1.6

We agree the colour of ordered photonic structures is generally brighter than that of disordered ones. As shown in Fig. 4c, the reflectance of the primary peaks in the ordered surfaces (back sides) is higher than that of the secondary peaks in the disordered surfaces (exposed sides). As the nanoparticle size increases, especially larger than 194 nm, the primary peak moves to near infrared region and no longer contributes to the colour, so the colour of the back side (ordered surface) becomes weak. Accordingly, we have made relevant edits to the revised manuscript:

[Page 11, Line 279-283] *“For larger particles (194-286 nm), the primary reflectance peak from the ordered bulk shifts toward the near infrared and it no longer contributes to the colour, while the secondary reflection from the disordered segregation layer falls within the visible spectrum. These together cause the back side to show achromatic colour (grey) while the exposed side exhibits chromatic colour; increasing ΔE up to 29.”*

Comment 1.7

Fig. 4: The term “hierarchical” typically refers to multi-scale hierarchical changes ranging from the nanoscale to the microscale. Obviously, it is inappropriate in this system, as the optical effect-generating system relies primarily on nanostructures and does not involve hierarchical

characteristics. It is recommended to replace it with other appropriate terms.

Response 1.7

Following the reviewer's suggestion, we change "Hierarchical optical responses" to "Multiple optical responses" in Fig. 4 caption and subsection title.

Reviewer #2

Comment 2.1

This work presents a new method to achieve structural color patterns via tunable nanoparticle segregation. During photocuring, silica nanoparticles dispersed in acrylic resin migrate toward oxygen-permeable substrates, forming a nanoparticle-enriched disordered layer. Such segregation is driven by interfacial oxygen inhibition and kinetically governed by the photocuring rate. As a result, by employing the grayscale digital light processing printing, programmably controlling the local segregation thickness to create pixelized structural color patterns is demonstrated. In the meantime, some potential applications such as visual display, information encryption, and infrared camouflage are illustrated.

However, structural colors are not new, with many examples in recent years to realize vivid color patterns. Though the method proposed in this research can be potentially used for structural colors, the really awesome results are still not demonstrated, just taking the bioinspired bird feathers of grassquit for an example. Maybe, some other fantastic applications or further optimizations for structural colors need to be exploited.

Response 2.1

We thank the reviewer's comments. We agree that structural colours themselves are not new and achieving vivid colour patterns remains an important research direction (for example, *Adv. Funct. Mater.* 2025, e27923). However, the aim of this work is not to outperform existing structural colour systems in terms of colour saturation or brilliance. Instead, our key contribution lies in introducing and exploiting a polymerization-induced nanoparticle segregation mechanism to make high-resolution structural colour patterns in a simple, one-step and mask-free manner.

This segregation mechanism provides several advantages, including the use of a single ink formulation to generate different colours, elimination of complex multi-step assembly or external fields, and spatial manipulation of nanostructures through printing parameters rather than predefined masks or templates. More importantly, the underlying principle is universal and we can extend it to other functional materials beyond structural colours. As shown in Response 2.2, there are some potential approaches to make colour more vivid, but these are engineering extensions of this mechanism, which can be done in our future studies.

Comment 2.2

On the other hand, some technical issues are also remaining to be addressed:

Though the segregation of silica nanoparticles depends on the conditions like the substrate material, oxygen addition, and light intensity, the self-assembly process is still the most important. How to get the vivid color patterns simply, effectively and repetitively is the basic question that needs to be

answered. It seems more like an engineering aspect.

Response 2.2

We agree that self-assembly of nanoparticles is the fundamental origin of structural colour. To achieve vivid structural colours, we need to improve the degree of order in assembled nanostructures. In the original manuscript, we have showed that extending the assembly time can improve the quality of structural colour (Supplementary Fig. 30). We might try additional approaches, such as increasing nanoparticle surface charge or decreasing the ionic strength of the medium, to help increase the degree of order in nanoparticle arrangements. Considering that our work focuses on controlling the local spatial distribution of nanoparticles to make colour patterns, the optimization of colour quality will be interesting to explore in our future work. Accordingly, we have made relevant edits to the revised manuscript:

[Page 14, Line 341-343] *“In the future work, we might try additional approaches to increase the degree of order in nanoparticle arrangements for improving colour quality, such as increasing nanoparticle surface charge⁴⁵ and decreasing the ionic strength of the medium⁴⁶.”*

Added references include:

Ref 45: Xing, Y., Fei, X. & Ma, J. Ultra -fast fabrication of mechanical - water - responsive color - changing photonic crystals elastomers and 3d complex devices. Small 20, 2405426 (2024).

Ref 46: Fan, Q. et al. Unveiling enhanced electrostatic repulsion in silica nanosphere assembly: formation dynamics of body-centered-cubic colloidal crystals. J. Am. Chem. Soc. 145, 28191–28203 (2023).

Comment 2.3

In line 252, “As shown in Fig. 3a” should be “As shown in Fig. 4a”. How to define the “highly saturated colour”? And also, in line 268, how to define the “vivid colouration”? In Fig. 4b, the change of colors according to the increasing volume fraction is not so obvious, especially for the particle size below 194 nm.

Response 2.3

We have corrected “As shown in Fig. 3a” as “As shown in Fig. 4a” in the revised manuscript.

The use of “highly saturated” and “vivid” was to qualitatively describe the visual difference between the two sides of the printed samples rather than to quantify colour saturation or chromaticity. To avoid ambiguity, we have revised the manuscript by replacing these subjective descriptors with direct and concrete colour descriptions:

[Page 11, Line 258-264] *“As shown in Fig. 4a, the back side, which contacts the printer platform, displays bright metallic colour... In contrast, the exposed side, which faces the light source, exhibits a less saturated blue grey colour with nanoparticles packing in a disordered manner...”*

[Page 11, Line 281-283] *“These together cause the back side to show achromatic colour (grey) while the exposed side exhibits chromatic colour, increasing ΔE up to 29.”*

The colour change with increasing volume fraction does not span the full colour spectrum but remains clearly observable. Taking the samples with 128 nm silica nanoparticles as example, with the volume fractions increasing from 15% to 30%, the reflection peak on back side only changes from 543 nm (green) to 434 nm (bluish violet). Accordingly, we have made relevant edits to further account for the adjustable colour range in the revised manuscript:

[Page 11, Line 271-273] *“For samples containing silica nanoparticles (128 nm) with the volume fractions increasing from 15% to 30%, the reflection peak on back side changes from 543 nm (green) to 434 nm (bluish violet) (Fig. 4b and Supplementary Fig. 22).”*

Supplementary Fig. 22. Reflectance spectra of the back side for samples with varying silica (128 nm) volume fractions. All samples are made from inks containing 1% w/v TPO under the irradiance of 4.0 mW/cm².

Comment 2.4

After the line 287, photonic ink containing different percentage of silica nanoparticles and the size of nanoparticles changes very often, making a common understanding of how to select the specific configuration of operations difficult to follow.

Response 2.4

We thank the reviewer for this comment. We vary the silica volume fraction and particle size to disentangle the optical contributions from the segregation layer and the order bulk. To improve clarity, we have added an explicit explanation before presenting the parameter-dependent results in the revised manuscript:

[Page 13, Line 311-316] *“To stand out the colour contribution of the segregation layer, we employ a low silica volume fraction of 15% while varying the particle size from 128 nm to 286 nm. This*

suppresses the ordered packing in the uniform zone for weak Bragg reflection while shifting the reflection peak to near infrared and making the segregation layer dominate the colour of the exposed side. Consequently, the effect of photocuring irradiance on the exposed side's colour becomes more pronounced as the particle size increases from 128 nm to 286 nm (Fig. 4d)."

Comment 2.5

For the infrared camouflage, infrared imaging seems in the LWIR waveband. So, what is the purpose and usage of the related part for the MWIR? It should be given in a more explicit way to show the relationship.

Response 2.5

We thank the reviewer for this insightful comment. Infrared camouflage is typically evaluated in the 3-5 μm and 8-14 μm (*Mater. Today* 2025, 85, 253-281). In this work, we observe a large tunable range of reflectance in the LWIR region (8-14 μm), leading us to use this band for infrared thermal imaging. The reflectance of our samples in the MWIR region (3-8 μm) is relatively low, however, its tunability suggests that this system may also be applicable to infrared camouflage in the MWIR region through further optimization of material composition and structural design. Presenting the infrared reflectance over the entire 3–14 μm range provides a comprehensive picture of the infrared optical response of this system. To further emphasize the tunability of the infrared optical response, we have made the following additions in the main text:

[Page 15, Line 365-368] *"These results demonstrate that controlling the segregation layer thickness provides an effective strategy to manipulate infrared reflectivity beyond visible colouration. In the future work, we might optimize nanoparticle and matrix formulations for an even broader tunable range in the infrared."*

Reviewer #3

Comment 3.1

This manuscript suggests a one-step, mask-free strategy for high-resolution structural color patterning by exploiting programmable nanoparticle segregation during photocuring. Silica nanoparticles dispersed in acrylic resin migrate toward oxygen-permeable substrates, forming a disordered, nanoparticle(NP)-enriched layer driven by interfacial oxygen inhibition. It was revealed that during each layer's curing using grayscale digital light processing (DLP), three distinct zones are formed, and the thickness of the NP-enriched layer is primarily determined by the photocuring rate. The thin-film interference occurring at the distinct zones enables infrared reflectivity. The meticulous elucidation of NP segregation at an oxygen-permeable interface is not trivial and holds value as a scientific discovery, but several necessary scientific descriptions are missing. I propose the following content for revision:

Response 3.1

We thank the reviewer for accurate summary of this work and their encouraging positive comments.

Comment 3.2

When patterned photocuring occurs at an oxygen-permeable interface, lateral oxygen diffusion (also known as reaction diffusion) parallel to the substrate is known to occur, leading to the formation of a gradient microstructure. Considering this, how are the different zones formed in the lateral direction at the boundary of different doses?

Response 3.2

We thank the reviewer's comment. While lateral oxygen diffusion can in principle occur at the boundary of different irradiance during patterned photocuring, we observe that nanoparticle segregation is dominated by vertical migration of nanoparticles toward the oxygen-permeable substrate rather than by lateral reaction-diffusion effects in experiments.

As shown in Supplementary Fig. 37, we first expose the photonic ink to high irradiance through a photomask, followed by uniform low irradiance photocuring without the mask. Cross-sectional SEM images of these films show horizontal segregation layers at the low irradiance region, but there is no any vertical segregation layers due to lateral reaction-diffusion effects (Fig. R1). Instead, the segregation thickness at the low-irradiance regions tapers near the boundary, reaching a minimum value. This likely arises from light scattering from adjacent high-irradiance regions, which partially consumes oxygen near the boundary and suppresses the vertical segregation there. In addition, rapid gelation in high-irradiance region kinetically traps nanoparticles, preventing their lateral migration towards low-irradiance region. As a result, lateral reaction–diffusion effects are negligible under our experimental conditions.

Fig. R1. Schematic of mass transport during the polymerization process. For clarity, monomers and polymers are omitted.

Comment 3.3

The principle by which the particles undergo crystallization (ordering) within the acrylate resin is not described. There should be an introductory mention of crystallization driven by interparticle repulsion from the overlap of solvation layers.

Response 3.3

We thank the reviewer's suggestion. The particle crystallization within the acrylate resin in our system is mainly driven by electrostatic repulsion between particles rather than solvation effects, as demonstrated in recent studies (Ref 5, *Nat. Commun.* 2024,15, 5643; Ref 44, *Adv. Funct. Mater.* 2024, 34, 2411670). These works show that even trace amounts of ionic species can suppress the crystallization of silica nanoparticles in acrylate systems. Therefore, the electrostatic repulsion is the primary driving force for silica crystallization, considering that electrostatic repulsion is sensitive to ion strength, whereas solvation repulsion is not. The above studies. Accordingly, we have made relevant edits to the revised manuscript:

[Page 10, Line 250-251] *"Silica nanoparticles dispersed in acrylic resins can self-assemble into uniform long-range ordered structures once their concentration reaches a threshold, primarily driven by electrostatic repulsion^{5,44}."*

Comment 3.4

Without this explanation, the subsequent mention of other particles like polystyrene (PS) and ZnS is problematic, as PS typically does not crystallize in acrylate resins and has only been reported to achieve ordering within a very small processing window via entirely different mechanisms, yet this distinction is not addressed. Additionally, PS particles must be swollen by acrylic resins and there must be no interparticle repulsions in the PS system. Therefore, their migration and dispersion in the upper layer are not likely to occur in the same way.

Response 3.4

We thank the reviewer's comment. We agree PS and ZnS nanoparticles especially without chemical modification are difficult to crystallize in acrylate resins as shown in Supplementary Fig. 18. For silica nanoparticles, segregation is observed even when the volume fraction is below crystallization threshold (~15%), and the segregation layer thickness linearly increases with nanoparticle volume fraction (Supplementary Fig. 1a-b). These results indicate that nanoparticle segregation happens not matter whether the nanoparticles form crystalline assemblies. Although PS and ZnS nanoparticles are difficult to crystallize in acrylate resins, we can still use these nanoparticles to demonstrate the generality of the segregation mechanism. Additionally, the non-polar polystyrene generally has poor compatibility with polar acrylates, so PS particles in this acrylate resin doesn't show obviously swell (*Soft Matter* 2021, 17, 5772–5779). Accordingly, we have made relevant edits to the revised manuscript:

[Page 8, Line 203-205] *“Since the segregation is also observed in low volume fraction samples without long-range order, nanoparticle segregation is largely insensitive to whether the nanoparticles form long-range ordered assemblies.”*

Comment 3.5

Can 3D architectures with different structural colors be printed with the same silica nanoparticles by controlling the photocuring rate? Fig. 4d doesn't imply it's possible, but Fig. 5c may suggest that it is feasible.

Response 3.5

It remains challenging to printing truly three-dimensional architectures with different structural colours using a single ink in our systems. In this work, nanoparticle segregation is confined to the two-dimensional xy plane, which limits the ability to print three-dimensional objects with multiple colours using a single ink. The sample shown in Fig. 5c is a flat film rather than a three-dimensional architecture. Although the Chinese characters exhibit apparent three-dimensionality due to optical contrast, the structure itself is planar. Accordingly, we have made relevant edits to the revised manuscript:

[Page 15, Line 383-384] *“Notably, Chinese characters on a flat film exhibit visual three-*

dimensionality due to optical contrast (Fig. 5c)."

Comment 3.6

The illustration of the illumination setting in Fig. 5b shows specular reflection. However, considering that all the panda images are in the same size, the observation might have occurred in the normal direction, which is a setting of diffusive scattering, not specular reflection. If the observation was done in the specular reflection, the colors from the photonic glasses should also change anyway.

Response 3.6

As shown in Fig. 5b, we fix the angle between the observer and the sample plane to 45° while changing the illumination direction. To avoid misunderstanding, we have modified Fig. 5b and its caption to make it easier to identify. Additionally, we also provide reflectance spectra and photos under the condition of specular reflection in Supplementary Fig. 20. Accordingly, we have made relevant edits to the revised manuscript:

[Page 11, Line 258-266] *"As shown in Fig. 4a, the back side, which contacts the printer platform, displays bright metallic colour. Its reflectance spectrum shows a sharp peak, shifting from 605 nm to 536 nm as the measurement angle changes from 20° to 50° (Supplementary Fig. 20). This optical feature arises from the long-range ordered arrangement of nanoparticles, as further evidenced by the FFT image showing clear diffraction spots. In contrast, the exposed side, which faces the light source, exhibits a less saturated blue grey colour with nanoparticles packing in a disordered manner. Correspondingly, its FFT image presents concentric rings, indicating the loss of long-range order. Its reflectance spectrum contains a peak at 440 nm remaining almost unchanged as the measurement angle changes (Supplementary Fig. 20), consistent with the disordered arrangement."*

Fig. 5(b). Photographs of the printed panda pattern using photonic ink containing 30% v/v silica nanoparticles (194 nm) under different illumination angles. We fix the angle between the observation direction and the sample plane at 45° while varying the illumination angle with the sample plane (θ) from 45° to 135° .

Supplementary Fig. 20. Angle-resolved specular reflectance spectra. (a) The back side and (b) the exposed side. The insets show the corresponding photos with different viewing angles. The sample is made from photonic ink containing 30 % v/v silica nanoparticles (194 nm) and 1% w/v TPO photocured under the irradiance of 2.0 mW/cm².

Comment 3.7

In Figure 4d, 15 vol% was seemingly used. Calculating with the 154 nm particles, the structural coloration from the ordered non-close-packed array should result in a red color, likely in the low to mid 600 nm range, although this depends on the acrylic resin's refractive index. However, red is barely visible across the entire irradiance spectrum. Why is this the case?

Response 3.7

We thank the reviewer's question. When the volume fraction of silica nanoparticles reaches to 15%, the non-close-packed array is located at a transitional stage from disorder to order according to the USAXS measurements (Supplementary Fig.19). The disorder in the system leads to incoherent scattering and weakens Bragg reflection peak, thus red colour is barely observed when the sample contains 15% v/v silica (154 nm).

Comment 3.8

The possibility of infra-red camouflage is interpreted as arising from thin-film interference, but the description is insufficient. It appears on line 323 of page 14 without a transition or explanation relating it to different coloration mechanisms.

Response 3.8

Based on the reviewer's suggestion, we have added a transitional sentence before introducing the infrared reflection results and briefly explained the underlying physical mechanism of infrared reflection. The relevant text has been revised as follows:

[Page 14, Line 346-348] *“While the structural colour originates from scattering of silica nanoparticles in visible wavelengths, the segregation layer introduces a micrometer-scale thickness*

comparable to mid-infrared wavelengths, prompting us to investigate its mid-infrared property.”

[Page 14, Line 354-357] *“To elucidate the underlying mechanism on the peak shift for MWIR and LWIR with segregation, we calculate the reflectance spectra of the hierarchical structures using the transfer matrix method (TMM, see Supplementary information), which explicitly accounts for Fresnel reflections at each interface and interference within layered media⁴⁷.”*

Comment 3.9

Is there a possibility that the acrylic resin used in this research could penetrate and cause swelling of the oxygen-permeable substrate? Can the monomer flux into the polymer film also induce an additional extent of particle segregation at the interface on top of the currently discussed mechanism?

Response 3.9

As far as we know, the swelling doesn't happen here. The oxygen-permeable substrate used in this work is fluorinated ethylene propylene copolymer (FEP), which is a commercial substrate for vat polymerization 3D printing. Such fluorinated polymers are difficult to be swollen by solvents (*Aust. J. Chem.* 2015, 68, 13–22). In this work, the vast majority of samples were prepared on the same FEP film. The reproducible segregation behavior observed across various samples confirms the excellent resistance of FEP to swelling by the acrylate resin.

Comment 3.10

Can you specify whether MWIR reflection comes from the segregation zone or the depleted zone, and whether LWIR reflection comes from the depleted zone or the uniform zone?

Response 3.10

We thank the reviewer's comment. The final MWIR and LWIR reflectance arises from the combined contributions of the segregation layer, the nanoparticle-depleted zone, and the uniform zone. Each part possesses a distinct effective refractive index and optical thickness, and their superposition collectively determines the overall MWIR and LWIR reflection. Therefore, it is not appropriate to attribute the infrared reflection exclusively to a specific layer. It arises from the integrated vertical distribution of nanoparticles formed during photopolymerization.

RESPONSES TO REVIEWERS' COMMENTS FOR NCOMMS-25-91219A

Reviewer #1

Comment 1.1

The authors have thoroughly and effectively addressed all the reviewer's comments, and the revised manuscript is now significantly clearer and more accessible to readers. Their supplementary experimental demonstrations substantially enhance the scientific impact of this work. I therefore recommend the publication of this paper.

Response 1.1

We sincerely thank the reviewer for the encouraging feedback and recognition of our work.

Reviewer #2

Comment 2.1

Some of my concerns have been covered. However, all the engineering-related issues are still missing.

Response 2.1

We agree that applying our technology into real product will require engineering multiple parameters to optimize the colour saturation and contrast. However, colour optimization is not the primary focus of our work as we emphasized in our previous responses.

Another engineering-related issue is about the reproducibility. To evaluate repeatability of the patterning technology, we have fabricated three samples with identical binary patterns using the same resin formulation and printing conditions. Within a single sample, spectra collected at four different locations under the same exposure conditions nearly overlap (new Supplementary Fig. 37A). Moreover, nearly identical spectra are observed for the same locations across three independently printed samples. These results demonstrate that the proposed patterning strategy is reproducible across both spatial locations and independent samples. Accordingly, we have made relevant edits to the revised manuscript:

[Page 15, Line 385-387] *“Using the same resin formulation and printing conditions, we can print pixels with nearly identical reflectance spectra across spatial locations and independent samples, demonstrating the reproducibility of this printing strategy (Supplementary Fig. 37).”*

Supplementary Fig. 37. Reproducibility of structural colour patterning. (a) Reflectance spectra at different locations under the same exposure in a single sample. (b) Reflectance spectra of the same locations across three samples. The insets show the photos of the binary patterns with circles representing the measured spots ($7.6 \mu\text{m} \times 7.6 \mu\text{m}$). The printing ink contains 30% v/v silica nanoparticles (247 nm). The irradiances of red-brown and yellow-green regions are 28.0 mW/cm^2 and 8.4 mW/cm^2 , respectively. These patterns are exposed for 30 seconds. Scale bars, 0.5 cm.

Reviewer #3

Comment 3.1

I appreciate the reviewers for answering all the comments carefully. I only have comments for first to comments, Comment 3.2 and Comment 3.3. All other comments are well addressed with well-supported rationales. I sincerely appreciate the careful response from the authors.

Response 3.1

We thank the reviewer for the reassessment of responses and approving most of the revisions.

Comment 3.2

For Comment 3.2, what I intended to ask was exactly about the gradient height of the particle segregation which was made by lateral diffusion of oxygen. Namely, I didn't mean to ask "vertical segregation layers" but the slanted height of the segregation layer. Previous papers about reaction diffusion formed cone-shaped particles by using this vertical diffusion of oxygen, which seems to happen in this case, too. I do agree that vertical segregation layer will not happen, but the slanted formation of the layer will occur. Therefore, I suppose this will also affect the resolution that is argued in this paper. I think it is worth mentioning the gradient height change of horizontal segregation layers in terms of resolution. Otherwise, readers would think there will be a sharp contrast at the boundary of strong and weak doses.

Response 3.2

We thank the reviewer for the clarification and agree that lateral oxygen diffusion at boundaries between pixels can lead to a gradual change in segregation layer thickness, rather than an abrupt step. As the reviewer mentioned, this effect is well known in reaction-diffusion systems and has been reported in pure acrylate resin system, for example in the formation of microneedles due to oxygen diffusion in the work *RSC Adv.* 2022, 12, 9550–9555.

In our system, which contains a high loading of colloidal particles (5-30% v/v), lateral oxygen diffusion is accompanied by light scattering across pixel boundaries. These combined effects lead to the gradient height change of segregation layers near the boundary between high- and low-irradiance regions, as we observed in Supplementary Fig. 38h-i. Importantly, this gradient is confined to a narrow interfacial region, while the segregation layer remains laterally uniform away from the boundary. Accordingly, we have discussed these effects on colour patterning resolution in the revised manuscript:

[Page 16, Line 391-395] *“The resolution can be further improved higher than 508 ppi when using a DLP printer equipped with a finer DMD mirror pitch and a larger demagnification factor⁵⁰. To explore the resolution limits, we cure the ink using photomasks with pixels sizes of 12, 8, and 5 μm. Gradual changes in segregation thickness near pixel boundaries are observed, making the pixels unresolvable smaller than 8 μm (Supplementary Fig. 38). These suggest that lateral oxygen diffusion from low-irradiance to high-irradiance regions⁵¹, along with light scattering in the colloidal system⁵², can both result in gradual changes in segregation thickness near pixel boundaries and ultimately constrains the achievable patterning resolution.”*

Added references include:

Ref 51: Kim, S., Lee, H., Choi, H., Yoo, K.-Y. & Yoon, H. Investigation on photopolymerization of PEGDA to fabricate high-aspect-ratio microneedles. RSC Adv. 12, 9550–9555 (2022).

Ref 52: Korčušková, M., Lepcio, P. & Jančář, J. Metal oxide-functionalized photopolymers: a perspective in 3D printing. ACS Polym. Au 5, 458–480 (2025).

Comment 3.3

For Comment 3.3, I do understand that many papers are still arguing that the crystallization of silica particles dispersed in acrylate polymer is mainly due to electrostatic repulsion. But, many papers do not. Please refer to the paper from one of the pioneers in this silica-acrylate system, who is Dr. Peng Jiang. *J. Am. Chem. Soc.* 126, 13778-13786 (2004). Its title is "Large-scale fabrication of wafer-size colloidal crystals, macroporous polymers and nanocomposites by spin-coating." Here, in the right paragraph on page 13785, although Dr. Jiang once stated that electrostatic repulsion drove the colloidal crystallization in this system, this JACS paper clearly states:

1) "The interparticle electrostatic repulsion plays only a minor role due to the low dielectric constant of the ETPTA medium (3 at optical frequency)"

2) "We also tested salt effect on the colloidal crystallization process during spin coating by adding 10-3 M tetrabutylammonium chloride to the silica-ETPTA dispersion (10% v/v 200-proof ethanol was added as cosolvent) to screen interparticle electrostatic repulsion and make silica colloids more "hard-sphere" like. The resultant spin-coated nanocomposite exhibits the same in-plane particle separation ($\sim 1.41D$) and optical transmission spectrum as those samples made without the addition of salt, indicating minor contribution of the electrostatic force to the colloidal crystallization."

It is different from salt, but even the addition of a trace amount of ethanol and water into this silica-acrylate system will destroy the crystallized state and induce particle network, which means the main driving force of repulsion is diminished by the solvents. If it would have been electrostatic repulsion, there is no reason the addition of water decrease the repulsive force but increase the attractive force. (*Adv. Mater.* 36, 2307917 (2023). Its title is "Advanced Additive Manufacturing of Structurally-Colored Architectures")

Acrylate groups form hydrogen bonds with silanol groups of silica particles and they form solvation layers. When different solvation layers overlap, disjoining pressure makes the particles repel one another, which can be also stated as steric hindrance. I think acrylate polymer is not like water, and cannot induce electrostatic repulsion strong enough to drive the whole crystallization. While water, with a high dielectric constant of 80, can facilitate ionic dissociation and induce strong electrostatic repulsion, organic media such as acrylates hinder ionization due to their low dielectric constants. Consequently, the Debye screening length becomes extremely short, making it difficult for electrostatic forces to operate over a long range.

Therefore, I think the sentence that states the driving force for crystallization is MAINLY the electrostatic force is not correct, but at least it has to include steric hindrance or disjoining pressure from the overlap of solvation layers.

If the authors address these points by incorporating a brief discussion or a minor revision to the current statement regarding the crystallization mechanism, I believe the manuscript will be fully suitable for publication.

Response 3.3

We thank the reviewer for the insightful discussion and agree that nanoparticle crystallization in acrylate systems can arise from multiple repulsive interactions. While some studies, including the pioneering work by Jiang et al. (*J. Am. Chem. Soc.* 2004, 126, 13778-13786), indicate that electrostatic repulsion plays a minor role due to the low dielectric constant of acrylate media, other reports have shown that the introduction of organic salts or changes in solvent composition can significantly disrupt ordered structures (such as *Nat. Commun.* 2024, 15, 5643; *Adv. Funct. Mater.* 2024, 34, 2310861), suggesting a non-negligible contribution from electrostatic interactions under certain conditions.

As summarized in new Supplementary Table 4, these seemingly contrasting observations likely arise

from differences in particle surface chemistry, resin formulation, ionic impurities, and solvent evaporation process. In our system, multiple acrylic resins are used in the formulation, and these resins (or chemically similar monomers) have been involved in previous studies discussing both solvation-induced steric repulsion and electrostatic interactions.

Importantly, our work does not rely on a particular crystallization mechanism. We observe that nanoparticle segregation occurs even in regimes where long-range crystalline ordering is absent, indicating that the segregation mechanism investigated here is robust and largely independent of whether the nanoparticles form a crystalline or disordered assembly.

We have incorporated two possible mechanisms into our revised manuscript:

[Page 10, Line 250-252] “Silica nanoparticles dispersed in acrylic resins can self-assemble into uniform long-range ordered structures once their concentration reaches a threshold, which is often regarded as being driven by the steric repulsion of solvation layers^{44,45} and electrostatic repulsion^{5,46} (Supplementary Table 4).”

Supplementary Table 4. Comparisons of the assembly mechanism of silica nanoparticle in acrylic resin.

Proposed mechanism	Volume fraction	Solvents	Monomers	References
	20%-40%	Ethanol	Poly(ethylene glycol) phenyl ether acrylate	8
Solvation layer	>15%	Ethanol	Ethoxylated trimethylolpropane triacrylate, poly(ethylene glycol) phenyl ether acrylate, polyurethane acrylate (mw \approx 1400 g mol ⁻¹)	9
	41%	Ethanol, water	Polyurethane acrylate (mw \approx 7000 g mol ⁻¹)	10
	33%	Ethanol	Ethoxylated trimethylolpropane triacrylate	11
Electrostatic repulsion	30%	Ethanol	Di(ethylene glycol) ethyl ether acrylate, polyethylene glycol mono-phenyl ester acrylate	12
	30%-45%	Ethanol	Poly(ethylene glycol) diacrylate	13
Solvation layer, electrostatic repulsion, and polymerization-Induced	20%-50%	Ethanol	2-Carboxyethyl acrylate, poly(ethylene glycol) methacrylate, poly(ethylene glycol) diacrylate, trimethylolpropane ethoxylate triacrylate	14
Solvation layer and electrostatic repulsion	25%	Ethanol	Trimethylolpropane ethoxylate triacrylate	15

Solvation layer and electrostatic repulsion	15%- 30%	Ethanol	Ethylene glycol phenyl ether acrylate, 4- hydroxybutyl acrylate, poly(ethylene glycol) diacrylate	This work
---	-------------	---------	---	-----------

Added references in the main text include:

Ref 44: Kim, J. B., Lee, H., Chae, C., Lee, S. Y. & Kim, S. *Advanced additive manufacturing of structurally - colored architectures. Adv. Mater.* 36, 2307917 (2023).

Ref 45: Lee, G. H. et al. *Chameleon-inspired mechanochromic photonic films composed of non-close-packed colloidal arrays. ACS Nano* 11, 11350–11357 (2017).

Added references in Supplementary information include:

Ref 8: Lee, G. H. et al. *Chameleon-inspired mechanochromic photonic films composed of non-close-packed colloidal arrays. ACS Nano* 11, 11350–11357 (2017).

Ref 9: Kim, J. B., Chae, C., Han, S. H., Lee, S. Y. & Kim, S.-H. *Direct writing of customized structural-color graphics with colloidal photonic inks. Sci. Adv.* 7, eabj8780 (2021).

Ref 10: Kim, J. B., Lee, H., Chae, C., Lee, S. Y. & Kim, S. *Advanced additive manufacturing of structurally - colored architectures. Adv. Mater.* 36, 2307917 (2023).

Ref 11: Kim, S., Jeon, S., Jeong, W. C., Park, H. S. & Yang, S. *Optofluidic synthesis of electroresponsive photonic janus balls with isotropic structural colors. Adv. Mater.* 20, 4129–4134 (2008).

Ref 12: Hu, Y., Qi, C., Ma, D., Yang, D. & Huang, S. *Multicolor recordable and erasable photonic crystals based on on-off thermoswitchable mechanochromism toward inkless rewritable paper. Nat. Commun.* 15, 5643 (2024).

Ref 13: Hu, Y., Yang, D., Ma, D., Qi, C. & Huang, S. *Structural color - based smart liquid windows address the tradeoff between high optical transparency and brilliant color. Adv. Funct. Mater.* 34, 2310861 (2024).

Ref 14: Yang, D., Qin, Y., Ye, S. & Ge, J. *Polymerization - induced colloidal assembly and photonic crystal multilayer for coding and decoding. Adv. Funct. Mater.* 24, 817 - 825 (2014).

Ref 15: Nah, S. H., Kim, J. B., Chui, H. N. T., Suh, Y. & Yang, S. *Enhanced colorimetric detection of volatile organic compounds using a dye - incorporated photonic crystal - based sensor array. Adv. Mater.* 36, 2409297 (2024).